# Decentralised Trust Layers for the Web: Towards Transparent AI-Powered Platforms

Quazi Mamun
School of Computing, Mathematics and Engineering,
Charles Sturt University
Albury, NSW, Australia
qmamun@csu.edu.au

Md Rafiqul Islam
School of Computing, Mathematics and Engineering,
Charles Sturt University
Albury, NSW, Australia
mislam@csu.edu.au

## Abstract

Artificial Intelligence (AI) increasingly mediates web experiences—ranking, recommendation, advertising, and moderation—yet independent verification of AI behaviour and data provenance remains rare. We present a *Decentralised Trust Layer (DTL)* that turns transparency from policy into protocol. DTL anchors provenance claims, binds deployed model versions to signed metadata, and produces privacy-preserving *inference receipts* in an append-only transparency log inspired by certificate transparency. We formalise a threat model, provide a deployable protocol suite (provenance anchoring, model lineage registry, inference transparency, and decentralised audit sampling), and specify a fully reproducible evaluation protocol. To remain faithful to what can be executed without proprietary dependencies, our measurements focus on cryptographic and logging overheads and proof verification costs; results indicate microsecond-scale per-request costs for receipt generation and logarithmic proof sizes consistent with transparency-log theory. We discuss governance, privacy, and regulatory alignment and outline how DTL can be integrated into existing web standards and platform architectures.

## Keywords

decentralised trust, transparency logs, data provenance, AI auditing, verifiable credentials, web governance

## ACM Reference Format:

Quazi Mamun and Md Rafiqul Islam. 2026. Decentralised Trust Layers for the Web: Towards Transparent AI-Powered Platforms. In *Proceedings of the 2nd International Workshop on Transformative Insights in Multi-faceted Evaluation (TIME 2026), held in conjunction with The Web Conference 2026 (TheWebConf '26), April 2026, Dubai, United Arab Emirates.* ACM, New York, NY, USA, 10 pages. https://doi.org/10.1145/nnnnnnn.nnnnnnn

## 1 Introduction

The contemporary digital landscape is undergoing a fundamental restructuring as artificial intelligence (AI) transitions from a backend utility to a primary mediator of human experience. On the modern web, algorithmic systems now govern information retrieval, social discourse, and commercial recommendation with unprecedented autonomy [1, 2]. However, this evolution has been accompanied by a systemic erosion of trust in centralised web platforms, which often operate as opaque "black boxes" under the exclusive control of a few technology giants [3]. As AI increasingly assumes responsibility for consequential decisions in healthcare, finance, and governance, the lack of transparency regarding data provenance and algorithmic behaviour has created significant societal risks, including the amplification of systemic bias and the proliferation of sophisticated misinformation.

AI-powered web platforms shape information access and visibility through ranking, recommendation, advertising selection, and moderation. Trust in these systems has eroded due to centralised control of data and models, limited transparency into algorithmic decisions, and difficulties in independently verifying claims about deployment, training data, or policy changes [4]. Documentation artefacts such as model cards and dataset datasheets improve disclosure, but they do not guarantee that a deployed service matches its published artefacts nor that records are immutable [5, 6].

This paper asks a practical systems question: *What would it take for users, developers, auditors, and regulators to independently verify key facts about an AI-powered web service—its data provenance, the model version serving a decision, and whether a decision log was rewritten—without requiring privileged access to the platform?* We argue that transparency must be backed by cryptographic commitments, append-only logs, and verifiable identities that make equivocation detectable and accountability enforceable.

*Core idea.* We propose a *Decentralised Trust Layer (DTL)*: a modular trust substrate that can be deployed as a sidecar to AI services. DTL provides: (i) tamper-evident **provenance anchoring** for data and transformations, (ii) a **model registry** for signed lineage and disclosures, (iii) **inference transparency** via append-only logging of privacy-preserving receipts, and (iv) decentralised **governance and auditing** for sampling-based verification and dispute resolution. DTL adapts transparency-log design patterns from the Web PKI [7] to AI inference events, and aligns with W3C identity and provenance standards [8–10].

*Novelty and positioning.* DTL does not introduce new cryptographic primitives; instead, it *protocolises* AI transparency for web decisions by turning narrative disclosures into append-only, non-equivocating commitments with verifiable receipts and consistency proofs. The novelty lies in (i) integrating provenance, signed model/policy registries, and inference receipts into a sidecar deployable architecture, and (ii) specifying an evaluation protocol that third parties can execute without privileged platform access.

*Contributions.* **Contributions.** We make four contributions: **System architecture** for a decentralised trust layer spanning provenance, model lineage, inference receipts, and governance, designed for incremental adoption and standards interoperability; **Formal threat model and security analysis** capturing security goals (integrity, non-equivocation, accountability, privacy) and adversary capabilities, plus a mapping to protocol defences; **Protocol suite** including receipt commitments, transparency proofs, and audit sampling; we provide pseudocode for implementers and highlight configuration choices; and **Reproducible evaluation protocol and runnable measurements** focusing on what can be executed without proprietary access: receipt generation, Merkle batching, proof sizes, and verification times, with confidence intervals and reporting guidance.

## 2 Problem Setting and Requirements

### 2.1 Stakeholders and trust boundaries

DTL targets web ecosystems where decisions are made by a platform operator but must be trusted by others: **users** (affected by ranking and moderation), **developers** (building on platform APIs), **auditors** (third-party assessors and researchers), and **regulators** (requiring traceability, record-keeping, and accountability). The operator controls infrastructure, models, and logs; DTL assumes the operator is *not* fully trusted to faithfully retain or disclose records.

### 2.2 Use cases

We focus on web-native use cases with high accountability demands: **Recommendation and ranking:** provide verifiable evidence that a recommendation was produced by model version $v$ under policy $p$ at time $t$.; **Conversational AI services:** bind responses to a deployed model build and a policy configuration (e.g., safety filters), while preserving input privacy.; **Content moderation:** log moderation decisions with policy digests for later dispute resolution and longitudinal auditing.; **Data provenance for web mining pipelines:** track dataset lineage and transformations used to train/refresh models..

### 2.3 Design requirements

From these use cases, we derive requirements: **(R1) verifiability** by non-privileged parties, **(R2) non-equivocation** (no split-view histories), **(R3) privacy-by-default** with selective disclosure, **(R4) incremental deployability** (sidecar or gateway), **(R5) interoperability** with web standards (DID/VC, PROV), and **(R6) measurable overhead** suitable for production services.

## 3 Background and Related Work

**Provenance and documentation.** PROV provides a standard model for representing provenance as entities, activities, and agents [10]. Model cards and datasheets articulate ML documentation and intended-use disclosure. Responsible AI literature emphasises that disclosure alone is insufficient without organisational and technical accountability mechanisms [11, 12].

**Transparency logs.** Certificate Transparency introduced append-only Merkle logs, inclusion proofs, and consistency proofs to detect certificate mis-issuance and log equivocation at Internet scale

[7, 13]. DTL generalises this pattern to AI *inference receipts* and model/policy versioning.

**Supply chain and attestations.** in-toto and SLSA formalise software supply-chain provenance, verification, and tamper-evident attestations [14, 15]. ML systems inherit analogous supply-chain risks (data poisoning, artefact substitution, dependency compromise) [16, 17]. DTL borrows attestation principles but targets web AI decisions rather than binaries alone.

**Decentralised identity and claims.** W3C DIDs and verifiable credentials provide decentralised identifiers and cryptographically verifiable claims [8, 9]. DTL uses DIDs/VCs to bind model registries, auditors, and operators to accountable identities and to publish signed metadata with controlled disclosure.

**Decentralised auditing of LLMs.** Recent work explores decentralised frameworks for auditing LLM reasoning [18]. Our goal differs: DTL focuses on *web platform transparency* across data provenance, model lineage, and inference events, and designs for incremental adoption via a sidecar architecture.

Table 1 summarises how DTL differs from adjacent approaches: it combines non-equivocating transparency logs with explicit bindings to model and policy versions, producing privacy-preserving inference receipts that enable third-party verification.

## 4 DTL Architecture

DTL is deployable as a sidecar microservice adjacent to model serving. Large artefacts (datasets, scripts, model cards) are stored off-chain (e.g., IPFS, Solid pods, or object stores) with content-addressed references [19]. Only compact commitments (hashes, signatures, roots) are anchored in an append-only structure or ledger.

### 4.1 Layer 1: provenance anchoring

DTL represents lineage using PROV: data artefacts as entities, transformations as activities, and organisations or tools as agents [10]. Each record contains: (i) a content hash of the artefact, (ii) a PROV statement (or reference), (iii) an agent identifier (DID), and (iv) a signature. Anchors may be posted to a ledger or a transparency log for tamper-evidence; the key property is append-only verifiable history, not the specific chain.

### 4.2 Layer 2: model registry

The registry binds a deployed model version $v$ to a metadata bundle including: model card, training data summary, evaluation report, dependencies and build provenance (SLSA-style statements), and policy-compatible constraints. The bundle is content-addressed and signed by the operator (and optionally third-party evaluators). Verifiers can check that a claimed version maps to an immutable bundle and that signatures correspond to known identities.

### 4.3 Layer 3: inference transparency log

DTL logs *inference receipts* per request (or per decision event). Receipts are hashed and appended as leaves in a Merkle tree, periodically publishing signed roots. Inclusion and consistency proofs allow any verifier to check a receipt is in the log and that the log has not been rewritten [7].

**Table 1: Positioning of DTL relative to adjacent transparency and provenance mechanisms (∼ denotes partial/optional support).**

| Approach | Non-equiv. log | Model bind. | Policy/decision bind. | Sidecar deployable | Auditor receipts/proofs | Privacy-by-default |
|---|---|---|---|---|---|---|
| PROV / model cards / datasheets | – | – | – | – | – | ∼ |
| Certificate Transparency-style logs | ✓ | – | – | – | ✓ | – |
| in-toto / SLSA attestations | ∼ | ∼ | – | – | ∼ | ∼ |
| DTL (this work) | ✓ | ✓ | ✓ | ✓ | ✓ | ✓ |

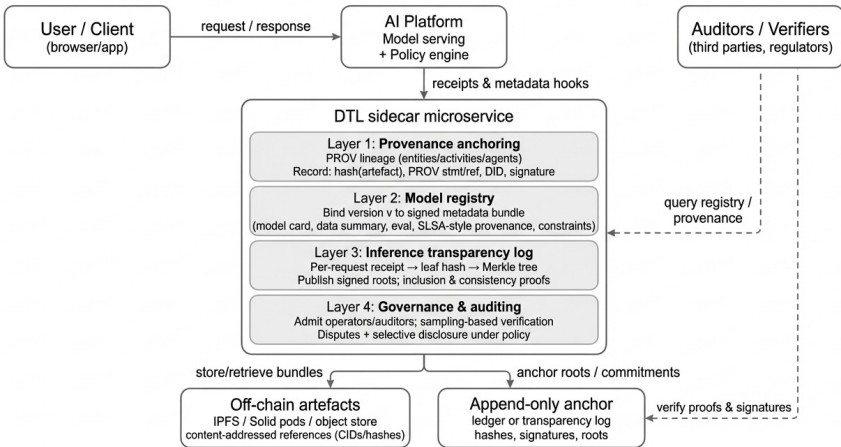

**(a) DTL deployment & trust flows (sidecar, off-chain artefacts, append-only anchoring, verification).**

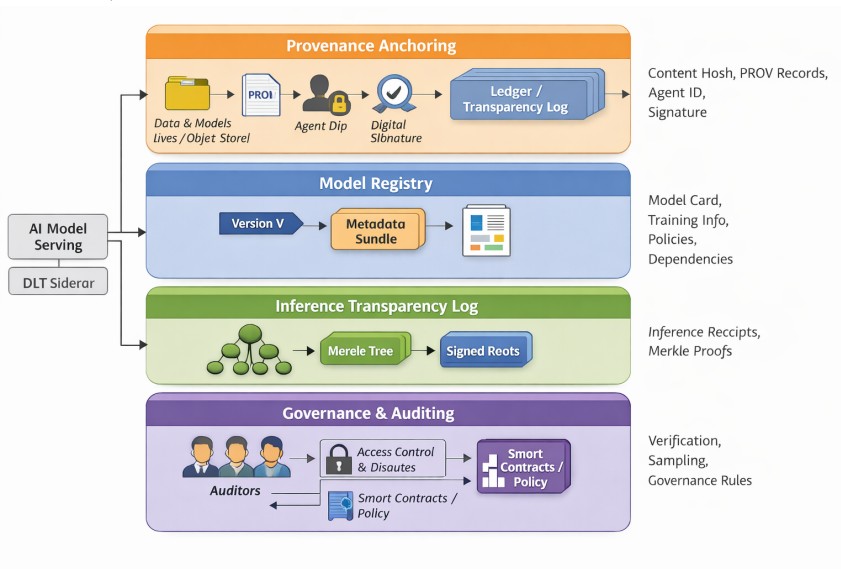

**(b) Layered DTL functions (provenance, registry, transparency log, governance).**

**Figure 1: DTL architecture overview.** *Top (a):* DTL is deployable as a sidecar adjacent to model serving; large artefacts (datasets, scripts, model cards) remain off-chain (e.g., object stores or content-addressed storage) and are referenced by content hashes, while compact commitments are appended to an append-only transparency structure. *Bottom (b):* Layered functions—provenance anchoring, model lineage registry, inference transparency, and governance—produce signed receipts plus inclusion/consistency proofs, enabling independent auditors to replay and verify web-facing decisions without access to proprietary internals.

## 4.4 Layer 4: governance and auditing

Governance defines who may operate logs, how auditors are admitted, and how disputes are resolved. Auditors sample receipts, verify proofs and signatures, and can request selective disclosure for contested cases. Governance policies can be codified as smart contracts or as a consortium policy; DTL remains agnostic, but requires that governance decisions are themselves recorded with accountable identities.

## 5 Protocol Suite

### 5.1 Receipt commitments and privacy

For an input $x$ and random salt $r$, define an input commitment:

$$c_x = H(x \| r). \tag{1}$$

Let the platform output be $y = f_v(x)$ and let $p$ denote the policy configuration. We define a minimal receipt:

$$\rho = (c_x, H(y), v, H(p), t, \sigma_{\text{op}}), \tag{2}$$

where $\sigma_{\text{op}}$ is a signature by the operator and $t$ is a timestamp. The log leaf is $L = H(\rho)$. Commitments enable later selective disclosure: a user can reveal $(x, r)$ to prove $c_x$ corresponds to an interaction without requiring the operator to expose raw inputs by default.

### 5.2 Merkle transparency and non-equivocation

Let $T_n$ be a Merkle tree with $n$ leaves and root $R_n$. A transparency log publishes signed roots $\sigma_{\text{log}}(R_n, n, t)$. Verifiers obtain: *inclusion proofs* (a leaf is in a given tree) and *consistency proofs* (a later tree extends an earlier tree) [7]. These proofs make split-view attacks detectable: if a log presents inconsistent histories, it cannot provide valid consistency proofs to all verifiers.

### 5.3 Audit sampling and detection

If auditors sample $k$ receipts from a window containing misbehaviour fraction $q$, the probability of detecting at least one violation under uniform sampling is:

$$P_{\text{detect}} = 1 - (1 - q)^k. \tag{3}$$

This yields a simple policy knob: increasing $k$ increases detection probability but raises audit cost. We later visualise Eq. 3 as part of the reproducible evaluation.

### 5.4 Dispute resolution (high-level)

A dispute bundles evidence: (i) the receipt $\rho$, (ii) inclusion/consistency proofs, (iii) the model registry entry for $v$, (iv) the policy digest $H(p)$, and optionally (v) selective disclosure of $(x, r)$ or plaintext output $y$ under governance rules. The governance layer adjudicates according to policy (e.g., mislabelling, policy noncompliance, or version misreporting).

## 6 Threat Model and Security Analysis

### 6.1 Security goals

evDTL targets the following properties. extbfRecord integrity: provenance anchors, model metadata, and receipts are append-only and tamper-evident. extbfPublic verifiability: any verifier can validate proofs and signatures without privileged access. extbfNon-equivocation:

---

**Algorithm 1** Receipt generation and logging (DTL sidecar)

**Require:** Input $x$, model version $v$, policy $p$
1: $y \leftarrow \text{SERVEMODEL}(x, v, p)$
2: $r \leftarrow \{0, 1\}^\lambda$ uniformly at random
3: $c_x \leftarrow H(x \| r)$
4: $\rho \leftarrow (c_x, H(y), v, H(p), t, \sigma_{\text{op}})$
5: $L \leftarrow H(\rho)$
6: Append $L$ to transparency log; obtain signed root and proofs
7: **return** $(y, \rho, \pi_{\text{incl}})$ to client

---

**Table 2: Threats and protocol defences.**

| Threat | DTL defence |
|---|---|
| Omit or rewrite decision logs | Merkle transparency log with signed roots and consistency proofs [7]. |
| Claim wrong model version/policy | Receipt binds $(v, H(p))$; registry binds $v$ to signed metadata bundle. |
| Substitute provenance artefacts | Content addressing (hashes) + signatures over PROV statements [10]. |
| Split-view equivocation | Inconsistent histories fail global consistency verification; gossip among auditors detects divergence. |
| Privacy leakage from receipts | Commitments $H(x\|r)$ and $H(y)$; selective disclosure under governance policies. |
| Sybil auditor influence | Admission via verifiable credentials; governance rules for quorum and penalty policies [9]. |

---

a platform cannot present divergent histories to different verifiers without detection. extbfAccountability: decisions link to a model version and policy digest at a given time. extbfPrivacy preservation: raw user data is not exposed by default; disclosures are selective.

### 6.2 Adversary capabilities

We consider adversaries who may:

(1) **Misbehaving operator**: omit receipts, rewrite logs, misreport model versions, or launder provenance.
(2) **Network attacker**: replay or delay messages, substitute artefacts, or attempt downgrade attacks.
(3) **Sybil/colluding auditors**: create multiple auditor identities or collude to suppress findings.
(4) **Curious verifiers**: infer sensitive user data from receipts and metadata.

### 6.3 Mitigation mapping

Table 2 maps threats to defences. The central mechanism is *cryptographic accountability*: receipts, model bundles, and provenance claims are signed; history is append-only and verifiable; and equivocation requires producing inconsistent proofs.

## 6.4  Discussion: residual risks

DTL does not eliminate all harms. A malicious operator can still choose harmful policies or deploy biased models; DTL increases detectability and evidentiary support, but normative questions remain (e.g., what counts as "fair" ranking). In addition, selective disclosure policies must be carefully designed to avoid enabling targeted surveillance or privacy regressions.

*Scope and non-goals.* DTL strengthens *verifiability* and *attribution* but does not, by itself, ensure that a model or policy is beneficial, fair, or socially acceptable. A platform could deploy a "lawful but harmful" model or policy (e.g., consistently biased ranking) while still producing perfectly valid receipts and logs. DTL's role is to make such deployments *auditable and contestable* by binding decisions to the exact model and policy versions used; mitigating normative harms requires complementary governance processes (e.g., pre-deployment risk assessments, fairness/robustness testing, human oversight, and appeal mechanisms).

## 7  Implementation Sketch

We implement DTL as a sidecar microservice with three primary modules: **(i) receipt engine** (hashing, signing, commitment creation), **(ii) log engine** (Merkle batching, root signing, proof generation), and **(iii) registry connector** (fetch and verify model bundles). A minimal deployment can run the log engine locally and periodically anchor roots externally; more robust deployments use multiple log witnesses and auditor gossip.

*Deployment considerations.* DTL is designed for incremental adoption. In a Kubernetes setting, the receipt/log engine can run as a per-pod sidecar alongside the model server, sharing request identifiers via headers and emitting signed receipts back to the client or to an internal policy/audit service. Alternatively, DTL can be deployed at an API gateway (or service mesh filter) to front multiple model replicas while preserving a consistent logging key. Keys should be generated and stored in a hardened keystore (e.g., KMS/HSM-backed signing) with explicit rotation policy; key identifiers are embedded in receipts to support long-lived verification. To bound per-request overhead, DTL batches leaves and publishes signed Merkle roots on a configurable cadence (every $N$ requests or every $T$ seconds). Critically, if a receipt cannot be issued or an inclusion proof cannot be obtained within a defined timeout, the system should fail *detectably* (e.g., return an explicit "no-receipt" status) rather than silently accepting unverifiable decisions. Robust deployments additionally use multiple log witnesses and auditor gossip to detect equivocation.

*Semi-realistic deployment case study.* To ground DTL in an operational setting, we outline an integration with an open-source LLM-backed web API (e.g., a moderation or conversational endpoint) deployed on Kubernetes. The primary container serves inference (e.g., vLLM/TGI/Triton behind a lightweight HTTP/gRPC gateway), while the DTL sidecar performs: (i) receipt construction over request/response digests, (ii) signature generation under a rotating service key, and (iii) asynchronous append to a local log engine that periodically publishes Merkle roots to an external transparency service. Two deployment modes expose the key trade-off: *strict* mode returns only after an inclusion proof is available (higher tail latency, stronger synchrony), whereas *async* mode returns a receipt identifier immediately and provides inclusion proofs on-demand (lower latency, proofs may lag by a batching window). Operational knobs include batching size, anchoring cadence (seconds vs requests), timestamp bucketing, and DID rotation to reduce cross-event linkage risk while preserving auditability.

*API surface.* A minimal interface includes: `/receipt` (generate receipt for inference), `/append` (append leaf batch), `/proof/inclusion` and `/proof/consistency` (return proofs), and `/registry/resolve?v={v}` (resolve model version to bundle digest).

## 8  Evaluation: Reproducible Protocol

We rewrite evaluation as an executable protocol aligned with overhead artefact evaluation expectations and constrained to what can be run without proprietary models, platform logs, or on-chain dependencies.

## 8.1  Scope and what we measure

To remain faithful to what can be executed, we measure: (i) per-request receipt construction overhead, (ii) Merkle batching/root computation overhead, (iii) inclusion-proof size and verification time as log size scales, and (iv) analytic audit-detection trade-offs derived from Eq. (3). We explicitly do *not* claim end-to-end latency for (public) chains, proprietary recommenders, or closed-source LLM serving stacks.

## 8.2  Datasets

The evaluation uses two **synthetic but released** datasets designed to exercise the DTL protocol surfaces while avoiding privacy risks. All artefacts are generated deterministically from fixed seeds and released as CSV/JSONL alongside scripts.

*D1: Synthetic recommendation traces (DTL-SynthRec).* We generate a trace of request–response events that mimics a ranking/recommendation service. Each event yields a receipt as in Eq. (2).
**Construction.** Fix a random seed $s_{rec}$. Generate: (i) a catalogue of $M$ items, each with a $d$-dimensional embedding $\mathbf{e}_i \in \mathbb{R}^d$, (ii) $N$ sessions, each with a user embedding $\mathbf{u} \in \mathbb{R}^d$, and (iii) for each request, a candidate set of size $C$ sampled uniformly without replacement. Scores are computed deterministically as

$$\text{score}(i) = \mathbf{u}^\top \mathbf{e}_i,$$

and the output $y$ is the ordered top-$k$ list of item IDs with their scores (rounded to a fixed precision). This yields deterministic outputs for fixed $(s_{rec}, M, d, C, k)$.

**Default parameters used for reported results.** $N = 65536$ requests per trial (as in §8.6), $M = 10000$, $d = 16$, $C = 200$, $k = 10$. (These parameters affect the synthetic payload sizes only; DTL overhead measurements are dominated by hashing/Merkle operations.)

**Released format.**
We export `data/synthrec/events.jsonl`: one JSON object per request: `{req_id, session_id, x_bytes_len, y_bytes_len, model_v, policy_id, timestamp}`. Raw $x$ and $y$ are optional; by

default, we store only lengths and stable digests to avoid unnecessary disclosure; and data/synthrec/payloads/: optional synthetic plaintext payloads for full end-to-end replay (disabled by default).

For transparency, we provide checksums for released files (SHA-256) and a manifest listing generation parameters.

*D2: Synthetic moderation prompts (DTL-SynthMod).* We generate templated prompts and policy categories to emulate content-moderation decisions. Each decision produces a receipt whose policy digest is $H(p)$.

**Construction.** Fix seed $s_{mod}$. Define a finite set of policy categories (e.g., benign, spam, harassment, hate, misinformation) and a template bank per category (short natural-language patterns). A deterministic rule maps each generated prompt to a category label and a binary decision (allow/block). We hash the policy configuration string $p$ (category definitions + thresholds) to obtain $H(p)$.

**Default parameters.** $N = 65536$ decisions per trial, with category proportions fixed by the seed.

**Released format.** data/synthmod/events.jsonl includes {req_id, x_bytes_len, decision, policy_id, timestamp} and optionally synthetic plaintext prompts.

## 8.3 Baselines

The three baselines used:

We compare against three baselines: **B0 No logging:** output only; **B1 Centralised mutable log:** append receipt rows into SQLite (WAL mode) without Merkle proofs. Schema: (req_id TEXT PRIMARY KEY, t INT, v TEXT, hp BLOB, cx BLOB, hy BLOB); and **B2 Transparency log only:** run the Merkle log (Layer 3) without provenance/registry resolution to isolate cryptographic/logging overheads.

## 8.4 Metrics

The evaluation metrics include:

The evaluation metrics include **Per-request overhead ($\mu$s):** time to construct receipt $\rho$ and leaf $L = H(\rho)$; **Commit overhead ($\mu$s/leaf):** amortised Merkle root computation per leaf under batch size $b$; **Proof size (bytes):** inclusion proof length ×32 bytes for SHA-256 nodes; **Verification time ($\mu$s):** time to verify inclusion proof for a sampled receipt; **Coverage:** fraction of requests with verifiable receipts (target 1.0 unless the operator omits); **Build time (ms):** total time to build a Merkle tree for $N$ leaves (reported in Table 3); and **p95 latency:** report p95 for proof generation and verification (Table 3 already reports mean/p95).

## 8.5 Experimental setup

*Implementation under test.* We benchmark the DTL sidecar modules responsible for: hashing, commitment construction $c_x = H(x\|r)$, receipt serialisation, Merkle batching/root computation, inclusion proof generation, and inclusion proof verification. The implementation is single-process and single-threaded during timing to minimise scheduler noise.

*Hardware and OS (reference environment for reported tables/figures).* All numbers reported in Tables 2–3 and Figures 2–4 were obtained

on: **CPU:** 56 vCPU, GenuineIntel (family 6, model 85) under a virtualised environment. **Memory:** 4 GiB RAM available to the container. **OS:** Linux kernel 4.4.0 (x86_64), glibc 2.36.

*Software stack.* **Python:** 3.11.2. **Crypto/Hash:** OpenSSL 3.0.17-backed primitives where applicable; SHA-256 via hashlib. **Numerics/plots:** NumPy 1.24.0; Matplotlib 3.7.5 (for figures only).

*Timing method.* We use time.perf_counter_ns() and report $\mu$s. Each trial includes a warm-up phase (first 1,000 requests discarded) to mitigate cache and interpreter warm-up artefacts.

## 8.6 Experimental design, precision targets, and confidence intervals

We follow the existing design and make the reporting requirements explicit.

*Runs.* For each batch size $b \in \{1, 2, 4, 8, 16, 32, 64, 128\}$, run $R = 5$ independent trials, each with $N = 65536$ requests.

*Confidence intervals.* Report mean ± 95% confidence intervals via non-parametric bootstrap (10,000 resamples) over trial means.

*Precision / power reporting (microbenchmark context).* Because $N$ is large within each trial, the uncertainty is dominated by cross-trial variance. We therefore treat the $R$ trial means as the independent units for CI computation and report: (i) mean and 95% CI, and (ii) a *minimum detectable effect* (MDE) at 80% power under a normal approximation using the observed across-trial standard deviation (script reports this automatically). This avoids over-claiming statistical power by incorrectly treating per-request timings as i.i.d.

*Reproducibility controls.* Fix seeds ($s_{rec}, s_{mod}$), publish all scripts, and release raw CSV measurements, manifests, and environment metadata.

## 8.7 Reproduction steps

The protocol is reproduced by the following steps (all artefacts are included in the accompanying package):

(1) **Generate datasets:** python3 scripts/generate_datasets.py --out data/ -N 65536 --seed 42
(2) **Run microbenchmarks (DTL and baselines):** python3 dtl_microbench.py --dataset data/synthrec/events.jsonl --trials 5 --batch-sizes 1 2 4 8 16 32 64 128
(3) **Export tables/figures:** python3 scripts/aggregate_and_plot.py --in results/ --out plots/
(4) **Verify checksums:** sha256sum -c data/MANIFEST.sha256

## 8.8 Results 1: receipt generation + Merkle batching

Table 3 reports $\mu$s/receipt (mean and 95% CI) for receipt generation plus Merkle batching across batch sizes. Figure 2 shows that the per-receipt cryptographic overhead remains in the low microsecond range across all batch sizes, indicating that receipt construction and hashing are unlikely to dominate end-to-end serving latency. The curve is relatively flat, suggesting that batching primarily amortises Merkle-root computation rather than changing the cost of receipt formation itself. This supports DTL's design goal of deployment as a sidecar in interactive services: platforms can tune batching to

**Table 3: Receipt generation + Merkle batching microbenchmark (μs/receipt, mean and 95% CI).**

| Batch size $b$ | μs/receipt | 95% CI |
|---|---|---|
| 1 | 5.894 | [5.665, 6.123] |
| 2 | 5.984 | [5.840, 6.124] |
| 4 | 6.284 | [6.097, 6.500] |
| 8 | 6.362 | [6.231, 6.492] |
| 16 | 6.254 | [6.109, 6.348] |
| 32 | 6.303 | [6.185, 6.467] |
| 64 | 6.221 | [6.164, 6.293] |
| 128 | 6.229 | [6.146, 6.331] |

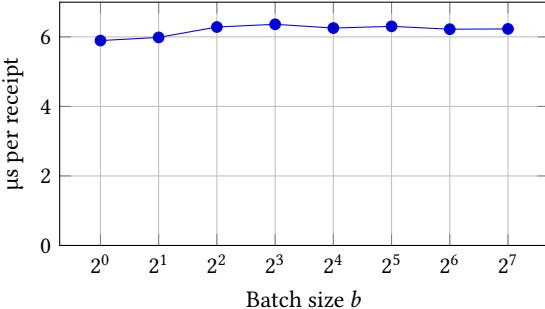

**Figure 2: Cryptographic/logging overhead for receipts and Merkle batching (reproducible microbenchmark).**

anchor cadence without materially affecting per-request receipt generation cost.

## 8.9 Results 2: proof size and verification scaling

DTL proofs are logarithmic in log size. We measure proof sizes and verification times as the number of leaves grows from $2^{10}$ to $2^{18}$. Table 4 shows proof bytes and verification time; Figure 3 illustrates the expected logarithmic scaling of inclusion proofs with tree depth: proof size increases linearly in the number of levels while remaining compact (hundreds of bytes) even for large logs. Verification time grows modestly with depth, consistent with the small number of hash operations required per proof. These trends imply that third-party auditing can remain lightweight: auditors can verify receipts at scale without needing access to platform internals, and proof transmission overhead stays small relative to typical web payload sizes.

## 8.10 Analytic audit trade-off visualisation

To support governance configuration, Figure 4 visualises the governance trade-off captured by Eq. (3): detection probability increases rapidly with the audit sample size $k$ when the misbehaviour rate $q$ is moderate, but grows more slowly for rare violations. This highlights why DTL couples transparency with governance: operational policies can choose $k$ to meet assurance targets (e.g., $P_{detect} \geq 0.95$) while controlling audit cost. In practice, this enables risk-based auditing, in which higher-stakes services or periods of elevated risk trigger higher sampling rates.

**Table 4: Merkle proof scaling (measured): $N$ leaves, depth, proof size, build time, proof generation and verification (mean/p95 in μs).**

| $N$ | depth | proof (B) | build (ms) | gen (μs) | verify (μs) |
|---|---|---|---|---|---|
| 1024 | 10 | 320 | 1.14 | 1.94/2.17 | 11.00/11.12 |
| 4096 | 12 | 384 | 38.04 | 2.64/3.83 | 14.54/21.04 |
| 16384 | 14 | 448 | 21.10 | 2.94/3.44 | 15.97/15.71 |
| 65536 | 16 | 512 | 89.15 | 3.99/4.79 | 17.91/18.28 |
| 262144 | 18 | 576 | 417.13 | 8.78/12.73 | 20.51/25.07 |

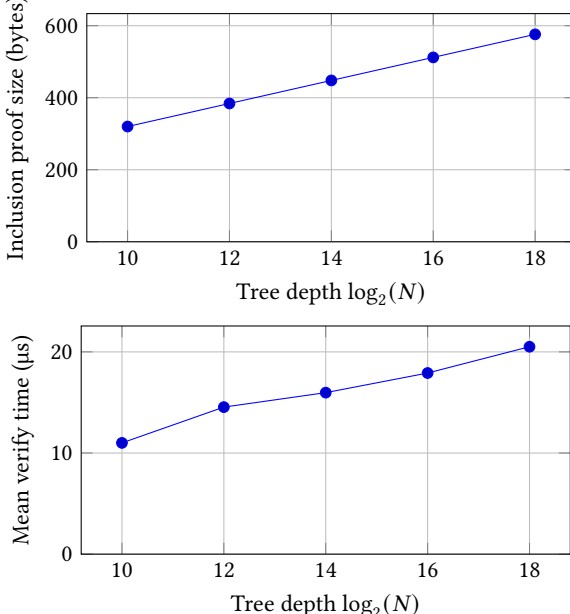

**Figure 3: Proof scaling: proof size grows linearly with depth; verification time grows modestly with depth.**

## 8.11 Extended experiment: end-to-end sidecar overhead in an open-source serving stack

This extended experiment is structured as a semi-realistic case study consistent with the integration described in Section 7. To complement the cryptographic microbenchmarks, we extend the evaluation protocol with an end-to-end experiment using an open-source serving stack that is reproducible by third parties. The goal is not model accuracy, but *system* impact: (i) p50/p95 request-latency overhead attributable to receipt issuance and proof availability, (ii) throughput impact under concurrent load, and (iii) storage growth per million receipts. A representative setup uses a containerised inference API (e.g., FastAPI/gRPC) with DTL as a co-located sidecar; a load generator issues requests with controlled concurrency and payload sizes. We recommend reporting end-to-end latency distributions (p50/p95/p99), sustained throughput, CPU utilisation, and log growth (bytes/request) under multiple batch sizes.

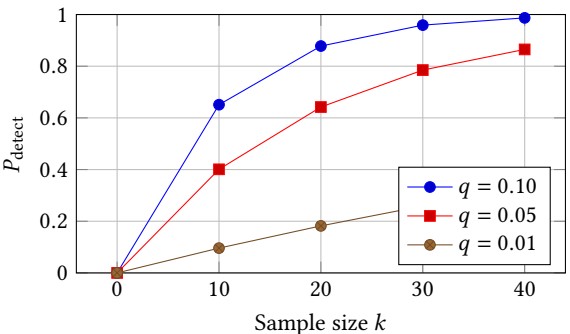

**Figure 4: Detection probability versus audit sample size under uniform sampling (Eq. 3).**

### 8.12 Extended experiment: anchoring cadence and failure-mode detectability

We further extend the protocol with (a) an anchoring-cadence study and (b) a failure-mode study to demonstrate *detectable* unverifiability. For cadence, vary root publication every $N$ requests versus every $T$ seconds, and report the trade-off between proof freshness and overhead (queueing/backpressure). For failures, inject controlled disruptions (e.g., temporary log outage, delayed root publication, or witness unavailability) and measure: (i) the rate of explicit "no-receipt" outcomes, (ii) the fraction of decisions lacking inclusion proofs within a timeout, and (iii) auditor detection outcomes (missing receipt, stale root, or inconsistency) as verifiable signals. These experiments connect the non-equivocation goals to operational behaviour and provide an adoption-relevant evidence trail without relying on proprietary platform access.

## 9 Discussion: Trade-offs, Interoperability, and Regulation

This section interprets the system and evaluation results in terms of deployment realities. We highlight the key engineering trade-offs (performance versus decentralisation), the privacy implications of verifiable logging, and how DTL can be adopted incrementally through standards-aligned interfaces. We also situate DTL as a technical substrate that can support compliance and independent assurance without prescribing a single governance regime.

### 9.1 Decentralisation vs performance

DTL is designed to decouple *what must be globally verifiable* from *what can remain local*. In practice, most deployments should treat the append-only anchor as a low-frequency commitment layer: receipts are batched, Merkle roots are published on a cadence aligned with operational needs (e.g., every minute or every $n$ requests), and only compact commitments (roots, signatures, digests) are anchored. This allows platforms to preserve interactive latency for end-users while still enabling later verification. The evaluation's microsecond-scale cryptographic overhead suggests that, for many services, receipt construction and hashing are not the dominant cost; rather, the anchoring cadence and proof distribution strategy determine user-perceived overhead.

A second performance consideration is the "verification surface". DTL does not require that *every* request be synchronously verified by third parties; instead, it supports asynchronous verification and sampling-based audits (Eq. 3). Operationally, this mirrors how certificate transparency is deployed: clients and monitors verify according to policy and capacity. For high-throughput systems, engineering choices such as witness replication, gossip frequency, and retention windows can be tuned to meet SLAs while preserving non-equivocation guarantees. Importantly, DTL's layering supports incremental adoption: deployments may start with Layer 3 receipts and add Layer 1/2 provenance and registry binding as maturity increases.

### 9.2 Privacy and selective disclosure

DTL's receipts are intentionally privacy-preserving by default, but privacy risks remain if identifiers or stable metadata enable linkage across sessions. Even when inputs are committed via $H(x\|r)$, repeated receipt fields (e.g., policy digests, model versions, coarse timestamps) can support correlation if not carefully scoped. A practical mitigation is to treat salts and identifiers as first-class privacy controls: salts should be high-entropy, per-request (or at least per-session), and never reused across contexts; timestamps can be bucketed when fine granularity is unnecessary; and any user identifiers should be avoided or replaced with unlinkable tokens.

Selective disclosure is also a governance risk: once a mechanism exists to reveal plaintext inputs/outputs, it must be constrained by an auditable policy [20]. DTL therefore benefits from "disclosure transparency" alongside inference transparency: access requests, approvals, and disclosures should themselves be logged (at least as commitments) so that an auditor can verify that disclosures occurred according to rules and were not silently over-used. In privacy-sensitive settings, deployments can further minimise exposure by disclosing only the minimal evidence required for a dispute (e.g., revealing $(x, r)$ without revealing unrelated context), and by enforcing strict retention limits consistent with data minimisation principles.

*Long-term privacy–verifiability trade-offs.* Even with salted commitments and selective disclosure, receipts and roots inevitably emit *metadata* (e.g., timestamps, service identifiers, model-version epochs, response sizes, and anchoring cadence). Over long horizons, these quasi-identifiers can enable cross-event linkage—especially when combined with external side information—even if payloads remain hidden. We therefore treat privacy as a tunable design dimension and add a systematic, auditor-facing risk assessment: choose a quasi-identifier set $Q$, apply bucketing/suppression, and estimate linkability via the expected anonymity-set size.

$$k_i(Q) = \left| \left\{ j : Q_j = Q_i \right\} \right|, \qquad L(Q) = \frac{1}{\mathbb{E}_i\left[k_i(Q)\right]}. \qquad (4)$$

Here, lower $L(Q)$ indicates less linkability (larger expected anonymity sets). Practical mitigations include (i) coarse time-bucketing, (ii) batching receipts/roots to blur individual events, (iii) rotating pseudonymous DIDs and signing keys, (iv) minimising or suppressing size and endpoint metadata, and (v) policy-driven retention and access controls. These mitigations trade off audit granularity, proof latency,

and investigative power; DTL's design makes this trade-off explicit and configurable rather than implicit.

## 9.3 Limits: lawful-but-harmful models and policies

DTL is intentionally *normative-agnostic*: it cannot prevent a platform from deploying a harmful yet compliant model or policy. What it can do is make such choices *transparent and attributable*. In particular, DTL can bind decisions to (i) model versions, (ii) declared policies (e.g., moderation/ranking rules), and (iii) signed evaluation artefacts (e.g., fairness audits, red-team reports, or safety assessments) expressed as verifiable credentials. This enables independent parties to verify whether a deployment followed a stated process and to contest outcomes with evidence. Preventing "lawful but harmful" behaviour requires complementary governance (e.g., enforceable policy constraints, pre-deployment thresholds, human review, and appeal/recourse mechanisms); DTL strengthens these by providing tamper-evident records that regulators and auditors can rely on.

## 9.4 Interoperability with web standards

A key design goal of DTL is to avoid introducing a bespoke ecosystem. Provenance anchoring can directly reuse PROV concepts (entities, activities, agents) and serialisations such as PROV-JSON or JSON-LD, which makes it feasible to integrate with existing data pipeline tooling and to exchange lineage claims across organisations [21]. Similarly, decentralised identifiers and verifiable credentials provide a standards-aligned way to represent accountable identities (operators, auditors, regulators) and to attach cryptographically verifiable claims to model metadata without requiring a single identity provider.

Interoperability also matters for adoption pathways: platforms already maintain artefact stores (object storage, registries, CI/CD logs) and can map these to DTL's content-addressed bundles rather than rebuilding infrastructure. In practice, many deployments can start by binding existing artefacts (model cards, evaluation reports, training summaries) into signed bundles and publishing their digests. Over time, the same interfaces can be extended to support multi-party attestations (e.g., an external evaluator co-signing a bundle) and cross-platform verification (e.g., auditors validating receipts against a shared transparency log format).

## 9.5 Regulatory alignment

Regulatory regimes increasingly emphasise traceability, record-keeping, and demonstrable accountability for AI-mediated decisions. DTL provides a concrete technical mechanism to support these requirements: model version binding makes it harder to "move the goalposts" after an incident; append-only logging preserves an evidentiary trail; and provenance anchors support claims about data sources and transformations. Importantly, DTL does not replace legal compliance processes, but it can strengthen them by making key facts independently verifiable rather than dependent on internal platform assertions.

DTL can also support proportionate governance. Not every domain requires the same transparency granularity; for example, consumer-facing recommendation may prioritise privacy and low overhead, while high-stakes domains may require stronger auditability and longer retention. By separating commitments (publicly verifiable) from bulk artefacts (selectively disclosed), DTL enables regulators and independent auditors to verify integrity without mandating full public disclosure of sensitive data or proprietary models. This "auditability without full openness" framing is often necessary to reconcile accountability with confidentiality and privacy constraints in real deployments.

*Operational guidance.* For production adoption, the most consequential knobs are key management and anchoring cadence. Signing keys should be managed via KMS/HSM-backed services with rotation and revocation semantics; receipts should embed key identifiers and policy/model digests to preserve long-term verifiability. Anchoring cadence ($N$ or $T$) trades proof freshness against overhead; deployments should choose defaults aligned with audit requirements (e.g., moderation decisions may require tighter freshness than low-stakes personalisation). Finally, DTL's security value depends on *detectable* failure behaviour: missing receipts, stale roots, and inconsistent proofs must be surfaced as verifiable conditions for auditors and, where appropriate, for end users.

## 10 Conclusion

AI-powered web platforms increasingly determine what information people see, how content is moderated, and which opportunities are surfaced through ranking and recommendation. As argued in the introduction, this shift has amplified longstanding concerns about centralised control, limited transparency, and the difficulty of independently verifying claims about model behaviour, policy changes, or data provenance. This paper addresses that gap by proposing the *Decentralised Trust Layer (DTL)*: a modular, standards-aligned trust substrate that can be deployed as a sidecar to existing model-serving stacks to make key accountability signals *verifiable by design* rather than dependent on platform disclosure.

DTL operationalises transparency through four complementary layers: (i) provenance anchoring using PROV-style lineage statements signed by accountable identities, (ii) a model registry that binds deployed versions to content-addressed and signed metadata bundles, (iii) an inference transparency log that produces privacy-preserving receipts and anchors Merkle roots in an append-only history, and (iv) governance and auditing mechanisms that enable sampling-based verification, dispute resolution, and selective disclosure under policy. Together, these layers provide a principled pathway to detect log rewriting and equivocation, to bind decisions to specific model and policy configurations, and to preserve evidence that can be independently checked by auditors and regulators. Our threat model clarifies the adversary capabilities DTL is designed to withstand and the security goals it prioritises (integrity, non-equivocation, accountability, and privacy). Finally, we translate evaluation into a reproducible protocol and report runnable microbenchmarks that establish a practical baseline for receipt construction and Merkle-based verification.

There are several directions for future work. First, while DTL's cryptographic primitives are lightweight, real-world deployments will require careful engineering around log witnessing, auditor gossip, and retention policies; evaluating these choices under realistic

traffic patterns and failure modes (e.g., partial outages, delayed anchoring, multi-region serving) is an important next step. Second, the privacy model can be strengthened by integrating selective disclosure mechanisms beyond salted commitments, including structured disclosure of receipt fields and privacy-preserving auditing workflows that reduce the need to reveal raw inputs or outputs during disputes. Third, governance remains a socio-technical challenge: future work should explore incentive-compatible auditor admission, Sybil resistance under decentralised identity, and mechanisms to make governance decisions themselves auditable without creating new central points of control. Fourth, broader interoperability studies are needed to map DTL bundles and receipts onto existing platform artefacts (CI/CD attestations, model evaluation pipelines, and content-policy systems), and to validate portability across multiple web platforms and jurisdictions.

DTL outlines a practical path towards a more trustworthy AI-mediated Web by combining provenance anchoring, version-bound model metadata, transparency-log-style inference receipts, and decentralised auditing. By shifting transparency from narrative claims to verifiable commitments, DTL supports user-centric accountability and provides a foundation upon which regulators, researchers, and platforms can build credible assurance processes as AI continues to reshape the digital public sphere.

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
