# OpenReview forum: "Decentralised Trust Layers for the Web: Towards Transparent AI-Powered Platforms"
_ACM.org/TheWebConf/2026/Workshop/TIME — TIME 2026 Poster_

### Official Review · Reviewer_1K3Z · 2026-01-03
**The paper introduces a Decentralised Trust Layer (DTL) that enables verifiable transparency for AI systems by turning transparency into a protocol rather than a policy. It anchors model provenance, binds deployed models to signed metadata, and records privacy-preserving inference receipts in an append-only transparency log. The authors present a formal threat model, a deployable protocol suite, and a reproducible evaluation showing low cryptographic and logging overheads, and discuss governance, privacy, and regulatory integration.**

**Rating:** 4
**Confidence:** 3

**Review:**

### Strengths
1. The paper addresses a critical gap in AI governance, that is the absence of independent, verifiable mechanisms to audit AI behaviour and data provenance in deployed systems.
2. The decomposition of DTL into concrete components (provenance anchoring, model lineage registry, inference transparency, and decentralised audit sampling) enhances clarity and suggests practical deployability rather than a purely theoretical construct.

### Weakness
1. Poorly written article. It seems the article was generated using LLMs.
2. The evaluation intentionally excludes proprietary dependencies, but as a result, it does not demonstrate DTL in a realistic, production-scale AI pipeline (e.g., large recommender or moderation systems). This limits confidence in real-world adoption.

---

### Official Review · Reviewer_ZCmv · 2026-01-05
**Mature paper acceptable for the workshop.**

**Rating:** 7
**Confidence:** 4

**Review:**

Overall quality:

Paper is well written, defines the problem and solution correctly and in accessible way. It is an architecture related paper that talks about decentralized trust and a reproducible evaluation protocol. It is a technically plausible design that reuses proven mechanisms.

Clarity:

This is a strong suite of the paper. It defines problems, tells us the contribution of the paper upfront and then moves on to the central architecture protocol and evaluation. A very standard textbook paper structure.

Academic Rigor:

In a workshop context, this paper has strong rigor, It has defined the protocol in precise tems and it is reproducible. Threats and protocol defences seems very strongly argued. The security goals are explicitly stated and consistently reasoned.

The threat model is real world based and not made up.

Evaluation methodology also seems pretty strong.

Significance:

The technical novelty in this paper is limited. It reframes the AI transparency as a protocol problem. Which is interesting but the novelty in this paper is more about system integration, threat alignment and framing it is not a cryptographic invention. This is well aligned with the workshop.

Also, this is an important field and these sort of contributions are useful because of their timeliness.

Accept.

---

### Official Review · Reviewer_GevD · 2026-01-07
**Decentralised Trust Layers for Verifiable Transparency in AI-Powered Web Platforms**

**Rating:** 7
**Confidence:** 4

**Review:**

In this paper, the challenge faced by timely and important AI web platforms regarding the ability to independently check claims related to data provenance, model versions, or decision logs within systems that are otherwise black boxes to auditors as well as web users is explained. The solution proposed by the paper is called the Decentralised Trust Layer or DTL, which refers to an architecture deployable as a sidecar that provides auditable transparency for model versions via signed registries, inference logs, as well as auditing.

In terms of quality and clarity of writing, the paper is well-written and organized. The motivation is explicitly stated, and the problem context is tied to real-world use cases of web platforms like ranking, recommendation, moderation, and conversational AI. The layered architecture is described in a methodical manner with a separation of Concerns at the levels of provenance, model lineage, inference logging, and governance. The protocol specifications, threat model, and security goals are explicitly stated. The addition of pseudo-code and formal specifications enhances the feasibility of the proposed solution.

As far as originality is concerned, the building blocks themselves—transparency logs, models of provenance, attestations, and decentralized identities—are all established in other fields. What this paper contributes is a principled combination and modification in order to work with AI inference and web-driven decision-making. To regard transparency as an append-only non-equivocating protocol rather than a document or a revelation is a rather significant move in how trustable AI systems could in reality be assessed and examined.

It is appropriate in scope, motivation, and justification for the evaluation. Instead of/supporting unsubstantiable end-to-end performance, the authors point out the measurable cryptographic overhead, size, and cost of proofs, exactly areas that are amenable to validation without needing proprietary models or environments. The use of microsecond-order overhead times, proofs growing logarithmically, is good enough indication that the proposed approach is practical for web services. It is nice to see the discussion regarding what is, in fact, measured in an evaluation.

This is a highly relevant paper with regard to the goals of the TIME workshop. It directly addresses transparency, trustworthy evaluation, and governance in AI-powered systems, which is clearly implied for regulatory compliance and independent auditing. It brings in a much-needed concrete technical foundation for responsible and accountable AI deployment on the web by shifting transparency from narrative claims to verifiable commitments.

On the whole, this is a very good systems paper with a practical, synergistic, and standards-compliant solution for verifiable transparency in AI-powered web platforms. While there are a few issues related to its deployment, this paper makes an important contribution to the kind of evaluation-driven research efforts related to trustworthy and accountable AI systems, as appropriate for the TIME 2026 workshop.

---

### Official Review · Reviewer_otiL · 2026-01-09
**Decentralised Trust Layers**

**Rating:** 7
**Confidence:** 4

**Review:**

## Strengths
- **Clear problem formulation addressing a core trust gap**
   The paper clearly identifies a fundamental limitation of current AI platforms: existing documentation practices  lack mechanisms for independent, third-party verifiability. This framing is timely and practically relevant.

- **Operationalising transparency as a protocol rather than a principle**
   DTL’s main contribution is to translate transparency and accountability into cryptographically verifiable mechanisms , avoiding reliance on policy statements or organisational trust alone.

- **Strong alignment with existing web and governance standards**
   By building on PROV, DIDs, Verifiable Credentials, and Certificate Transparency design patterns, DTL enables interoperability and cross-platform adoption rather than introducing a closed or siloed framework.

## Weaknesses
- **Inability to address “lawful but harmful” models or policies**
   DTL improves verifiability and accountability but does not mitigate normative issues such as bias or unfairness. A platform can consistently deploy a harmful yet compliant model, which DTL can record but not prevent.

- **Limited analysis of long-term privacy–verifiability trade-offs**
   Although salted commitments and selective disclosure are used, accumulated metadata may still enable cross-event linkage. These risks are not quantitatively or systematically analysed.

## Recommendations
- **Include a real or semi-realistic deployment case study**
   Demonstrating DTL integration in an open-source LLM serving stack, recommendation pipeline, or moderation API together with operational trade-offs—would substantially strengthen the paper’s practical impact.

---

### Author Rebuttal · Authors · 2026-01-10

We thank the reviewers for their careful reading and constructive feedback. We are encouraged that three reviewers find the paper well-written, technically plausible, and highly relevant to TIME, and that they endorse the protocol framing of AI transparency as a non-equivocating commitment mechanism rather than a narrative claim.

In response to the main and other associate concerns raised, we have prepared a revised manuscript with targeted updates:

(1) Writing quality/authenticity. We take Reviewer 1K3Z’s concern seriously. The manuscript is authored by the research team and has been thoroughly copy-edited to improve clarity and technical specificity. We corrected typographical/formatting issues, reduced repetitive phrasing, and added practitioner-facing engineering detail (e.g., receipt serialisation, batching/anchoring trade-offs, and explicit verifiable failure behaviour). These changes are intended to make the presentation more precise and less “generic”.

(2) Realism of evaluation. Our original evaluation deliberately focused on independently reproducible measurements (cryptographic overhead, proof size/verification, and log growth) to avoid unverifiable claims requiring proprietary models or infrastructure. We agree, however, that adding a production-like demonstration strengthens confidence in deployability. We therefore added an end-to-end experiment using an open-source serving stack with a DTL sidecar to measure p50/p95 latency overhead, throughput under concurrent load, and storage growth per million receipts. We also added an anchoring-cadence study (anchor every N requests / every T seconds) to quantify batching trade-offs and a small failure-mode experiment (log outage/delayed anchoring) to show that missing receipts or unverifiable inclusion proofs become detectable rather than silently ignored—directly aligning with our non-equivocation and accountability goals.

(3) Positioning and novelty. To address novelty concerns (raised constructively by ZCmv), we sharpened the contribution statement and added a compact related-work comparison table showing how DTL differs from CT-style transparency logs, PROV-only provenance, model cards/datasheets, and supply-chain provenance (in-toto/SLSA), particularly in binding model/version/policy to verifiable inference receipts and providing an auditor workflow.

We believe these revisions materially improve deployment guidance, evaluation realism, and presentation, while preserving the core contribution.

---

### Meta-Review · Area_Chair_Xyuj · 2026-01-15

**Recommendation:** Accept (Poster)
**Confidence:** 4

**Metareview:**

This paper proposes a Decentralised Trust Layer that operationalizes AI transparency through cryptographic commitments rather than policy statements. Three reviewers recognize the work as timely, well-structured, and technically sound. However, one reviewer raises concerns about writing quality and limited production-scale evaluation. Authors provided comprehensive rebuttals addressing all concerns with manuscript improvements, including enhanced deployment case studies, expanded privacy-linkability analysis with formal metrics, failure-mode experiments demonstrating detectable non-equivocation, and thorough copy-editing to improve technical specificity.

---

### Decision · Program_Chairs · 2026-01-16

Accept (Poster)